# An AI-Inspired Spatio-Temporal Neural Network for EEG-Based Emotional Status

**DOI:** 10.3390/s23010498

**Published:** 2023-01-02

**Authors:** Fahad Mazaed Alotaibi

**Affiliations:** 1Department of Information systems, Faculty of Computing and Information Technology (FCIT), King Abdulaziz University, Jeddah 22254, Saudi Arabia; 2College of Dentistry, Chosun University, Gwangju 61452, Republic of Korea

**Keywords:** brain–computer interfacing, human–robot interaction, EEG signal, deep neural networks, emotion recognition

## Abstract

The accurate identification of the human emotional status is crucial for an efficient human–robot interaction (HRI). As such, we have witnessed extensive research efforts made in developing robust and accurate brain–computer interfacing models based on diverse biosignals. In particular, previous research has shown that an Electroencephalogram (EEG) can provide deep insight into the state of emotion. Recently, various handcrafted and deep neural network (DNN) models were proposed by researchers for extracting emotion-relevant features, which offer limited robustness to noise that leads to reduced precision and increased computational complexity. The DNN models developed to date were shown to be efficient in extracting robust features relevant to emotion classification; however, their massive feature dimensionality problem leads to a high computational load. In this paper, we propose a bag-of-hybrid-deep-features (BoHDF) extraction model for classifying EEG signals into their respective emotion class. The invariance and robustness of the BoHDF is further enhanced by transforming EEG signals into 2D spectrograms before the feature extraction stage. Such a time-frequency representation fits well with the time-varying behavior of EEG patterns. Here, we propose to combine the deep features from the GoogLeNet fully connected layer (one of the simplest DNN models) together with the OMTLBP_SMC texture-based features, which we recently developed, followed by a K-nearest neighbor (KNN) clustering algorithm. The proposed model, when evaluated on the DEAP and SEED databases, achieves a 93.83 and 96.95% recognition accuracy, respectively. The experimental results using the proposed BoHDF-based algorithm show an improved performance in comparison to previously reported works with similar setups.

## 1. Introduction

In the last decade, research efforts, from both academia and industry, have shifted toward the development of human interactive social robots that can build socio-emotional relationships with humans. The human emotional status has a strong impact on the daily life decision-making process [1]. Social robots require a precise understanding of human emotion for effective assistance during daily routine tasks [2]. Social robots require a highly robust brain–computer interfacing (BCI) framework for the precise identification of the human emotional status [3]. To bring noise robustness and invariance in the identification of human emotional states, both physiological and non-physiological signals are employed either simultaneously or separately.

The non-physiological signals include facial expression [4], vocal pattern, and text data [5,6,7]. Emotional status identification via facial expressions requires uninterrupted facial cues within each frame of the video. The overall framework includes face detection, localization, feature extraction, and facial landmark tracking using a certain machine learning mechanism with adequate training data and a stable evaluation strategy [8]. Face expression-based emotion identification is a contactless method and can track multiple subjects simultaneously; however, the photometric variations in the personal appearance of the subject degrade the accuracy of the emotion identification model. The vocal pattern also varies with the emotional conditions of the well-being. Vocal pattern-dependent frameworks require stable and distinctive speech temporal signal representation and classification models. Vocal pattern-based methods are very prone to environmental noise. Text mining and natural language processing applications require identifying the subject’s emotions from the text presented [9]. The frequency of words is employed in the feature extraction mechanism for text-based emotion identification. Non-physiological signal-based emotion recognition techniques are less complex; however, they can easily be deceived if the subject hides their true facial expressions, vocal tone, and textual words selection. Moreover, the non-physiological signal-dependent emotion recognition techniques fail when the subject is suffering from physical impairments [10].

The physiological signals include skin impedance [11], respiration [12], and blood volume pulsation [13], among others. It is hard for the subject to conceal their emotions in physiological signals. In comparison to other biosignals, the EEG patterns collected from various scalp locations on the skull can describe the emotional pattern more effectively in comparison to other biosignals, such as an ECG, and EMG signals [14]. In the recent decade, various frameworks were developed for emotion identification based on EEG biosignals. However, the development of a robust and invariant features extraction system continues to be a challenging task. Channel selection and signal pre-processing always play a crucial role in the robustness of emotion identification systems. The unwanted noise such as moment artifacts, respiration, cardiac beats, and blood flow need to be removed to improve the classification accuracy of the human emotion identification frameworks [15,16]. A human–robot interaction requires a hardware device to sense the EEG signals, which spatially covers the scalp and senses the electrical activity of the neurons in the brain cortex. The multi-channel EEG devices consisting of numerous mini electrodes record the voltage amplitude values at a specified sampling frequency. The data collected by a multi-channel EEG device are pre-processed before the feature extraction and classification stage. The feature extraction model for EEG signals can either be based on handcrafted features or deep features [17]. The handcrafted features extraction stage is independent of the training procedure of the classifier. On the other hand, the deep neural network model requires a large amount of training data to adjust the weights of the neurons through an iterative optimization algorithm and combines both the feature extraction and the classification stages together. In many cases, existing neural network models are employed for the extraction of features through transfer learning to reduce the complexity and training stage with large training datasets.

In most existing applications, the pre-existing neural network models are employed for the extraction of features through transfer learning methods. The features are not only extracted directly from the raw one-dimensional EEG signals but also from the two-dimensional spectrogram of the EEG data. The pre-existing DNN models for feature extraction include ResNet, GoogLeNet, VGG-19, and InceptionV3, which can be combined with classification models, such as the multi-class SVM, KNN, and decision trees. Such hybrid models can work effectively on time patterns as well as spectrograms of EEG data. The emotion identification framework precision is hugely dependent upon the pre-processing methods, the robust feature extraction models, and the final classification models. However, the non-linear nature of the EEG results in large numbers of false positives and false negative cases which degrade the performance of the existing emotion identification frameworks. The PhysioNet repository publicly provides complex physiologic EEG signals for a happy, sad, and normal emotional status for subjects with varying gender and age. In this work, we approach the problem of emotion recognition from EEG data using a new hybrid framework. This contribution involves the fusion of both handcrafted and robust deep features using a novel bag-of-features (BoF) model for solving the emotion recognition problem from EEG biosignals. More specifically, the main contributions of this paper are as follows:A new hybrid feature extraction framework is developed for emotional status identification using a robust fusion approach of handcrafted features together with DNN-based features.The Short-Time Fourier Transform is used to map the 1D EEG patterns into 2D spectrograms before the final feature extraction stage.A bag-of-features framework is used to reduce the dimension of the hybrid raw feature vector.A KNN-based clustering approach is introduced to combine the feature vectors from multiple EEG channels.

The remainder of this paper is arranged as follows: Section 2 describes the existing literature related to the problems addressed in this research. Section 3 discusses the proposed methodology, Section 4 presents the different experimental results, and finally, Section 5 provides a conclusion and some future research directions.

## 2. Related Work

The traditional EEG-based emotion identification schemes were developed along three main directions: pre-processing, feature extraction, and classification. Moreover, feature selection and reduction are some of the post-processing methods used to lower the computational complexity and storage requirements of diverse systems. In the pre-processing stage, the EEG signals are conditioned using various filtering approaches to remove the unwanted noise artifacts. To induce the emotional status of subjects, various data types are used, such as images [18], textual data [19], music clips [20], and videos [21], with different degrees of success. In what follows, we briefly describe the main stages involved in the development of emotion identification frameworks from EEG data. We then present our proposed framework.

### 2.1. Data Pre-Processing

The pre-processing stage includes effective channel selection, the filtering of the raw EEGs, and 2D spectral representation. The channel selection criterion aims at reducing raw data by excluding the EEG channels with fewer interclass variations [22]. Researchers adopted various methods to pick the most effective EEGs out of 64 channels on the basis of statistical information. In [23], the statistical dependencies of the data collected by an electrode positioned at a different area of the scalp were used for the selection of channels. Figure 1 presents the configuration and spatial distribution of 64-channel EEG electrodes on the human scalp for recording the electrical activity of brain neurons. Optimal channels subset identification methods were broadly classified into two classes based on their dependency on the classification model. The classifier-independent method [24], relying on a threshold value of 1.725 on the differential entropy, was applied for the selection of effective channels in the SEED [25] and DEAP [26] datasets. A total of 12 channels in SEED and 32 channels in the DEAP database fulfilled the differential thresholding criterion. In [27], 10 optimal EEG channels out of 64 were identified by analyzing the statistical parameters, such as the Shannon entropy, differential entropy, and first difference method. In [28], the normalized mutual information was estimated based on inner channel connection matrix thresholding to identify an optimal EEG channel subset. In [29], mutual information maximization based on joint entropy was used to detect an optimal subset of the EEG channels. Both entropy and mutual information-based methods do not rely on the classification model; however, it degrades the accuracy of the overall model due to the absence of correlation details among the EEG channels [30]. The classifier-dependent methods developed in [31,32] select those EEG channels for which the classification performance is high. Classifier-dependent methods do not affect the accuracy of the overall model; however, they are sensitive to feature extraction methods [32].

In general, the data acquisition devices record diverse noise artifacts due to joint movements, blood flow, and inconsistent environmental noise. Both personal and environmental artifacts removal require carefully designed filters at the pre-processing stage of the proposed pattern identification framework. In [33], raw EEG signals were passed through a low-pass filter to exclude the unwanted artifacts. The EEG signals were de-noised and the continuous wavelet transform (CWT) was applied to further reduce the noise interference. The discrete wavelet transform was employed in [34] to identify and filter out unwanted frequency components from the raw EEGs. Learnable temporal filters were introduced in [35] that learn and optimize the filter weights and parameters for EEG-based brain activity pattern identification. Instead of raw EEGs, the spectrogram of an EEG using Short-Time Fourier Transform (STFT) [36] was employed as a pre-processing method to enhance the classification accuracy of the model. The hybrid deep lightweight features were extracted from both a 1D raw EEG and their 2D spectrogram using a 1D-LBP and pre-trained DNN, respectively [37]. The various pre-processing methods are summarized in Table 1.

### 2.2. Feature Extraction

The pre-processing stage is followed by the feature extraction stage. Both the 1D and 2D signal-based feature extraction models developed so far can broadly be classified into two classes, namely handcrafted and deep learning-based methods. The handcrafted features based on statistical parameters are manually engineered by data scientists, while the deep learning-based methods are automatically learned from the labeled data provided for neural network weights optimization. In [38], the power spectrum of a frequency-based segmented EEG denotes the elements of the feature vector. The differential asymmetry (DASM) [39] and asymmetry rational (RASM) [40] were computed from the difference in absolute power and the ratio of absolute power between symmetrical channel pairs. The power-based method is less complex and low dimensional; however, it provides a lower accuracy for emotion identification. Entropy-based features descriptors, including simple entropy [41], permutation entropy (PE) [42], spectral entropy (SE) [43], singular value decomposition entropy (SVDE) [44], and approximate entropy (ApEn) and sample entropy (SampEn) [45], were all developed to represent the non-linear patterns of the EEG signals. Time domain features have less complexity; however, the non-stationary nature of the EEG makes the time domain features less representative and leads to degradation in the accuracy. Diverse frequency domain features were developed to lower the noise impact of the EEG data acquisition procedures. Frequency domain features include the Power Spectral Density (PSD) [46] and Short-Time Fourier Transform (STFT) [47], among others, and provide a better spectral estimation for informative EEG segments. Although frequency domain features are more noise robust in comparison to time domain features, frequency domain features have a higher computational complexity and are hence less cost-effective. Time-frequency domain features provide high noise robustness when compared to frequency domain features.

The development of neural network models for data classification has also helped researchers in improving the categorization capability of their models. The LSTM was used in [48] to extract features directly for the EEG patterns and then to fuse these. The LSTM has a low complexity and can be applied directly to time domain EEG data. However, the LSTM showed a low performance in emotion recognition. In [49], EEGNet was employed with better compactness and complexity compared to the LSTM. The EEG data were represented in a 3D shape and classified with a 3D-CNN in [50]. The Pearson Correlation Coefficient was employed to shuffle the EEG channels before feature extraction in [51]. In [52], the EEG features were extracted with a recurrent neural network (RNN) in a parallel mode to improve the accuracy of the model. The 2D spectrogram version of the EEG signal was described with various pre-trained DNN models, such as AlexNet [53], ResNet [54], LeNet [55], VGGNet [56], and InceptionV3 [57], and a high accuracy was obtained; however, the dimensionality was extremely high for all 64 EEG channels. The deep neural networks require expensive GPUs for feature extraction, which is not affordable and thus limits the deployment in many emotion identification applications. The details of various feature extraction methods are summarized in Table 1.

### 2.3. Feature Selection and Transformation

Feature reduction methods aim at reducing the dimension of the feature vector by either feature selection or transformation. The resulting feature variables should have high interclass variations and low intraclass variability. Feature reduction methods identify appropriate features with a strong relationship to class labels. The raw extracted feature vectors are usually subject to feature reduction approaches to reduce the memory requirements and complexity. Many researchers have adopted correlation-based feature reduction methods such as the correlation-based Analysis of Variance (ANOVA) and Kendall’s rank coefficient. The learning methods recently developed include sparse regression, and LASSO uses L1 regularization for reducing the dimension of the feature vector. The stochastic methods depend on the global minima for feature variable selection. Robust feature reduction methods, such as Particle Swarm Optimization (PSO) [58], the Genetic Algorithm (GA) [59], Simulated Annealing (SA) algorithm, Correlation Feature Selection (CFS), and Minimum Redundancy Maximum Relevance (mRMR), outperformed conventional methods, such as the PCA, LDA, and ICA. A summary of common feature reduction/transformation algorithms is provided in Table 1.

### 2.4. Classification Algorithms

The classification stage is the last one in the EEG processing pipeline. The performance of a classifier is measured by comparing the true class label with the estimated one by the classifier. Researchers have employed both shallow learning classifiers and deep learning classifiers for emotion recognition. The shallow learning classifiers include the SVM, kNN, DT, MLP, and others, while the deep learning classifier includes the CNN, DNN, DBN, and other modified versions. In [60], the SEED and DEAP datasets were classified with the SVM and KNN, achieving 93.8 and 77.4% classification accuracy, respectively. The differential evolution-based channel selection algorithm (DEFS_Ch) [61] emotion classification model employed the linear discriminant analysis (LDA) and achieved 86.85% accuracy. The 2D transformation of an EEG was obtained with Azimuthal Equidistant Projection (AEP) [62] and employed the DNN model, achieving 96.09% accuracy. In [63], hybrid handcrafted features were used with an SVM achieving 75.64% accuracy. The CapsNet [64] classification on DEAP achieved 68.2% accuracy. In [65], the flexible analytical wavelet transform (FAWT) features from the SEED dataset gave 83.3% accuracy. The reduced set of features obtained through recursive emotion feature elimination (RFE) gave 60.5 and 90.33% using the DEAP and SEED datasets, respectively. A brief summary of the most common classification methods is given in Table 1.

### 2.5. EEG Datasets for Emotion Recognition

Human emotion identification can be visual based, oral based, and biosignal based, depending upon the application of interest. The visual-based emotion is identified through the facial expression and body posture of the subject. The oral-based emotion is identified with the pitch and volume of the subject’s voice. Both the visual- and oral-based emotion identification methods are less complex; however, the methods are less effective as subjects can conceal their real emotions in facial expressions and voice tones [66]. Therefore, biosignals have been considered as real effective sources for human emotion identification. The multi-channel electrodes are attached to the human scalp at defined positions to record the neural activity in the brain. EEG databases consisting of various emotion classes of different age subjects have been developed and are publicly available. The emotional state of the subject is excited with the help of different inputs, including audio, video, and images. The databases mostly employed in the emotional categorization experiments include GAMEEMO [67], K-EmoCon [68], SEED [69], ASCERTAIN [70], DEAP [26], MAHNOB-HCI [71], and IADS [72]. The summary of the above-mentioned databases is available in Table 2.

## 3. Proposed Workflow

In this section, we discuss the detailed architecture of our proposed workflow. The proposed model consists of a pre-processing stage and a spectrogram transformation stage, followed by the feature extraction and the classification stages, as shown in Figure 2. The filtering process is performed with a bandpass Butterworth filter designed for the specified range of frequencies. The spectrogram transformation stage converts the 1D EEG signals into 2D spectrograms. The raw feature vector is obtained by fusing deep features from the GoogLeNet DNN and the handcrafted OMTLBP_SMC [73] (developed by the authors). The feature clustering is performed to find the vocabulary vectors from the training set. The final hybrid feature vector is the frequency of raw features with similarities to the vocabulary vector set. To avoid classifier bias, a simple KNN classifier is then used to estimate the class label for the emotional state of the subject. The different stages of the proposed workflow are now discussed in more detail.

### 3.1. BoDHF Validation Datasets

The proposed BoDHF was validated and tested on SEED [69] and DEAP [26] datasets. The SEED dataset consists of 15 subjects in total from a set of 62 electrodes. A total of 3 experiments were performed in different sessions with 24 trials in each. The duration of the SEED EEG signal varies in duration from 185 to 265 s. The DEAP dataset was collected from 32 subjects over 32 EEG electrodes, with 1 experiment in each of the 40 trials. The EEG channel data of 63 s was recorded over a sampling rate of 128 Hz.

### 3.2. BoDHF Data Pre-Processing

Noise in raw EEG signals is present due to both internal and external artifacts. The internal artifacts are due to eyeball movement, eye blinking, heart pumping, respiration, and muscle movements. The external noise rise is mainly due to electrode displacement. To remove both lower and higher frequency noises, a simple bandpass Butterworth filter is used for the range of 4–45 Hz. Next, and because brain activity produces non-stationary EEG patterns with varying frequency content with time, the 1D filtered EEG signal is converted into a 2D spectrogram using the Short-Time Fourier Transform. Small segments of the EEG signal are obtained using overlapping windows to analyze the temporal variations in the spectrum. The generalized expression for the STFT is given in Equation (Equation 1).
(1)$$\begin{array}{c} S\left(m,k\right)=\sum_{i=1}^{L-1}E\left(i+f_{m}\tau \right)w\left(i\right)e^{\left(-j\frac{2\pi }{L}ik\right)} \end{array}$$
where S(m,k) denotes the *m*th component of the time-frequency frame of a spectrogram, k denotes the Fourier coefficient, L is the window length in samples i, w(i) is the *i*th sample of the overlapping window, and the $f_{m}$ is the shifting step of the overlap window. The output 2D spectrogram depends on the shape of the overlap window employed, such as Unimodal, Gaussian, and Symmetric. The time resolution of the spectrogram depends on the size of the overlapping window. A small window provides more time localization, while a large window decreases time resolution but increases frequency discrimination. The values at each index of time and frequency are represented by the amplitude of the spectrogram at that position. The expression for the amplitude is defined in Equation (Equation 2).
(2)$$\begin{array}{c} A\left(m,k\right)=\frac{1}{L}{\left|S\left(m,k\right)\right|}^{2} \end{array}$$
where A(m,k) represents the amplitude component of the spectrogram for a specified time-frequency frame. The spectrograms in Figure 3 show examples of the neutral, positive, and negative emotional states of a given subject.

### 3.3. BoDHF BoDHF Raw Feature Extraction

The proposed bag-of-hybrid-deep-features (BoHDF) extraction model consists of the pre-processing method followed by the raw features extraction stage. The raw feature vector is a fusion of the GoogLeNet fully connected layer feature values and the texture feature values obtained from the OMTLBP_SMC [73]. The hybrid combination of deep and handcrafted features extraction method combines the high-level convolutional features with a low-level pattern representation model to capture more distinctive interclass variations. The OMTLBP_SMC describes the pattern of micro-structures that exist inside the time-frequency image of each EEG signal. The OMTLBP_SMC is invariant to orientation and translational changes in the input data and provides robustness against noise artifacts. The OMTLBP_SMC adds distinctiveness to the neural features and enhances the performance of the overall model. The OMTLBP_SMC is extracted from a 3 × 3 patch of the spectrogram. The variable *P* shown in Figure 2 (feature extraction stage) denotes the overlapped multi-scale fused segment of the spectrogram.
(3)$$\begin{array}{c} P_{k}^{3}={\left\{\frac{\left(\sum_{i=0}^{6}\left(A_{0^{\circ }+15^{\circ }(i)+45^{\circ }(k-2)}^{3}\right)\right)}{7}\right\}}_{k=1}^{8} \end{array}$$
(4)$$\begin{array}{c} P_{k}^{2}={\left\{\frac{\left(\sum_{i=0}^{4}\left(A_{0^{\circ }+22.5^{\circ }(i)+45^{\circ }(k-2)}^{2}\right)\right)}{5}\right\}}_{k=1}^{8} \end{array}$$
(5)$$\begin{array}{c} P_{k}^{1}={\left\{A_{0^{\circ }+45^{\circ }(k-1)}^{1}\right\}}_{k=1}^{8} \end{array}$$
where the symbol *P* denotes the segment obtained through overlapped multi-oriented pixel fusion process of the spectrogram’s amplitude value at radius 2 and 3 around the position of the pixels as shown in Equations (Equation 3) and (Equation 4). The variable *i* defines the amplitude values lying at a specific radius, which gives the kth angle value in the multi-scale fused patch. Sample points $P_{k}^{3}$, $P_{k}^{2}$, and $P_{k}^{1}$ express the topological structure of 8 sample values having radius 3, 2, and 1, respectively. The multi-scale fused segment in Figure 2 is obtained by averaging the similar-oriented adjacent multi-oriented fused segment. The similar orientation values at radius 1 and 2 are fused together while the similar oriented at radius 2 and 3 are combined to obtain three topological structures shown in the proposed framework of the model. The multi-scale fusion is defined in Equation (Equation 6).
(6)$$\begin{array}{c} {\tilde{U}}_{v}={\left\{\frac{\sum_{i=(v-1)}^{v}\left(P_{d}^{i}\right)}{2}\right\}}_{d=1}^{8} \end{array}$$
where ${\tilde{U}}_{v}$ defines the multi-scale fused topological structure with the subscript *v* expressing the radius of the topological 8 structure shown in Figure 2. The $P_{d}^{i}$ in Equation (Equation 6) represents the $d^{th}$ orientation values of the segments’ multi-oriented fused patch.

The sign and magnitude components are obtained from ${\tilde{U}}_{v}$ using Local Difference Sign-Magnitude Transform (LDSMT) shown in Equations (Equation 7) and (Equation 8). In the third stage, the feature vectors are extracted from each topological structure as shown in Figure 2.
(7)$$\begin{array}{c} Q_{v}={\left\{{\tilde{U}}_{v}^{d}-C_{v}\right\}}_{d=1}^{P} \end{array}$$
(8)$$\begin{array}{c} Q_{v}={\left\{s_{v}^{d}\times m_{v}^{d}\right\}}_{d=1}^{P}\;\mathrm{and}\;\{\begin{array}{c} s_{v}^{d}=sign\left(Q_{v}^{d}\right)\\ m_{v}^{d}=\left|Q_{v}^{d}\right| \end{array} \end{array}$$
where $Q_{v}$ is the local difference value, ${\tilde{U}}_{v}^{d}$ is the *d*th sample point at radius *d*, and $C_{v}$ represents the center amplitude value corresponding to segment ${\tilde{U}}_{v}$. $s_{v}^{k}$ denotes the *d*th sign operator with a value equal to 1 when $d_{v}^{k}$ is larger than or equal to 0, and −1 otherwise. The $m_{v}$ represents the magnitude component, containing the absolute values of the $d_{v}$. The scale invariant operator OMTLBP_S${}_{P,R}^{riu2}$ is calculated by using the $s_{v}$ operator; the encoding of this is similar to LBP${}_{P,R}^{riu2}$. The *P* and *R* define the sample points and the radius values. The rotation invariant operator OMTLBP_M${}_{P,R}^{riu2}$ shown in Equation (Equation 9) is obtained from $d_{v}$.
(9)$$\begin{array}{c} OMTLBP\_M_{P,R}^{riu2}=\{\begin{array}{c} \eta \left(m_{v}^{d},\rho_{m}\right),\;\mathrm{if}\;\zeta_{P,R}\leq 2\\ P+1,\;\;\;\mathrm{otherwise} \end{array} \end{array}$$
(10)$$\begin{array}{c} \zeta_{P,R}=|\eta \left(m_{v}^{P-1},\rho_{m}\right)-\eta \left(m_{v}^{0},\rho_{m}\right)|+\sum_{d=1}^{P-1}\left|\eta \left(m_{v}^{d},\rho_{m}\right)-\eta \left(m_{v}^{d-1},\rho_{m}\right)\right| \end{array}$$
where η(i,j) is 1 when *i* larger than or equal to *j* and η(x) = 0 when i<j. The $m_{v}^{d}$ denotes the $d^{th}$ magnitude component, while the $\rho_{m}$ defines the average value obtained from the magnitude component of the segment at position(v) of the spectrogram. The OMTLBP_C is formulated as
(11)$$\begin{array}{c} OMTLBP\_C_{P,R}=\eta \left(C_{v},\rho_{I}\right) \end{array}$$
where $\rho_{I}$ is the average of the compete spectrogram. The joint combination of the three operators is expressed with OMTLBP_SMC${}_{P,R}^{riu2}$ describing the complete texture feature of the spectrogram. The OMTLBP_SMC [73] and GoogLeNet features have been concatenated at the feature stage of the proposed workflow as shown in Figure 2.

The DNN feature vector is obtained from the GoogLeNet model (other network structures were also considered). GoogLeNet is a multi-layer deep neural network model initially described in [74]. The 22-layer GoogLeNet has reduced complexity and was found not to overfit the EEG data signals, hence outperforming other existing DNN models in the classification of 2D transformed EEG signals. The GoogLeNet is almost 12 times fewer parameters than Alexnet. The network was developed for easy deployment over low-power electronics, such as mobile phones. Its basic idea is to convolve in parallel different sizes from most accurate (1 × 1) to bigger filters (5 × 5). Overall, the GoogLeNet was used for our application as it trains faster than VGG, and the size of a pre-trained network is comparatively smaller than VGG. Moreover, the nine inception layers of the GoogLeNet boost the deep learning outputs and avoid overfitting. For our application, the 2D spectrogram images are resized to 224×224 to match the input layer of the GoogLeNet model. The spectrogram is fed into the GoogLeNet and processed through the convolutional and max-pooling layers of the network. The pooling layer of the model reduces the 3×3 and 5×5 blocks into a single value. The rectification is performed with the traditional RELU expression given in Equation (Equation 12).
(12)$$ReLU=max\left(0,x\right)$$
where the ReLU is the rectified linear value, which reflects the maximum value between the input and 0. We have collected our desired feature values from the fully connected layer of the model, while the rest of the DNN stages were ignored. The resulting 1000 DNN features were concatenated to 600 OMTLBP feature vector to obtain the final raw feature vector of size 1600, as shown in Figure 2.

The feature extraction process is followed by a clustering stage. Here, the training data are used to create a vocabulary set. All extracted raw feature vectors are processed to create “*k*” groups of feature vectors. The k-clusters are created using an iterative process, described in Equations (Equation 13) and (Equation 14). The total of *k*-cluster centroids are initially selected, randomly represented with $\psi_{j}$, with *j* ranging from 1 to *k*.
(13)$$\psi_{j}=\frac{\sum_{i=1}^{m}\left(x_{j}^{i}\right)}{m}$$
(14)$$x_{j}=\arg min_{j}\left|\right|x^{i}-\psi_{j}{\left|\right|}^{2}$$
where $\psi_{j}$ denotes the centroid of cluster *j*, and the $x_{j}^{i}$ defines the *i*th feature vector belonging to cluster *j*. The iterative process continues until the difference between the centroid in the *i*th and i−1 iteration reaches a certain threshold or limit. The cluster centroids for each class describe the vocabulary set of the dataset. The final histogram is the frequencies of occurrence of each vocabulary centroid in the raw feature data representing the final feature vector. The bag-of-hybrid-deep-features (BoHDF) vector for any emotional status is extracted by calculating the frequency of vocabulary in the raw features of spectrograms. The test features are classified using a simple K-nearest neighbor (KNN) and support vector machine (SVM) classifiers. Such classifiers are chosen given their simplicity, satisfactory performance, and to check the overall performance of the proposed workflow independently of the power of the final classification stage.

## 4. Experimental Results and Discussion

For our experimental setup, 32 channels out of 64 EEG electrodes were selected for the emotion recognition problem. The hybrid features consisting of the handcrafted and deep features were used with various classifiers, including the ensemble, decision tree, SVM, and KNN with a Gaussian kernel, cubic, quadratic, and weighted Gaussian, to identify the best combination of classifiers and kernels. An Intel^®^Core i7-6500U CPU with 2.50 GHz and 8 GB RAM with MATLAB software was used to implement and test the performance of the proposed workflow. The proposed framework was evaluated on the publicly available PhysioNet’s DEAP and SEED datasets. The datasets were split into 70% training and 30% testing segments with a 5-fold cross-validation procedure to train and test the classifier(s).

The classification performance of the proposed model was evaluated using two publicly available databases, namely the SEED and DEAP datasets. The BoHDF model was used for the feature extraction stage while the SVM, ensemble, tree, and KNN classifiers were used to classify the test EEG patterns into their respective emotion classes. The DNN models tested in this research work include the GoogLeNet, AlexNet, ResNet-50, ResNet-101, Inceptionv3, and ResNet models. The images of dimension 224 × 224 (from the spectrograms) were used as inputs to the different network’s first layer. The row feature vectors of dimension 1024 were then extracted at the fully connected layer. Finally, the OMTLBP_SMC feature vector of dimension 600 was added for the complete description of the EEG patterns. Different cluster sizes were considered, k=8,10,12, to find the optimal value of k. The cluster size 8 provided the highest classification accuracy on both datasets presented in Table 3. The performance of the proposed model is compared with the Multiband Feature Matrix (MFM) [64], Multivariate Empirical Mode Decomposition (MEMD) [75], Mel-Frequency Cepstral Coefficients (MFCC) [76], Recursive Feature Elimination (RFE) [77], differential entropy (DE) [58], bag of deep features (BoDF) [60], Empirical Mode Decomposition (EMD) [78], Shannon’s entropy [79], EMD combined with sample entropy (EMDSE) [47], spatial-temporal recurrent neural network (STRNN) [65], and Higher-Order Statistics (HOS) [80], and the results show that the proposed workflow consistently outperforms the existing state-of-the-art approaches on both the DEAP and SEED databases. The proposed model, consisting of GoogLeNet fully connected layer features, when tested on both the DEAP and SEED datasets, outperformed the state-of-the-art, as mentioned in Table 3. The proposed model in integration with GoogLeNet and the KNN classifier achieved the best performance of 93.83% and 95.95% on the DEAP and SEED databases, respectively.

In Table 4, different clusters were used to extract the BoDHF features integrated with different DNN features and classifiers, and the classification performance of the GoogLeNet and KNN on cluster size 8 has achieved the highest accuracy. The classification performance of the ensemble classifier provides an accuracy of 94.6% with GoogLeNet with cluster size 8, which is higher than AlexNet, ResNet-50, ResNet-101, and InceptionV3 which gave an accuracy of 94.2, 93.3, 93.2, and 93.5%, respectively. The tree classifier provided the highest accuracy of up to 92.6% with GoogLeNet and the tree classifier and a cluster size of 8; however, this is lower than that obtained with cluster size 12. The cluster size 12 on GoogLeNet and the tree classifier provided a 94.2% accuracy which is higher than the AlexNet, ResNet-50, ResNet-101, and InceptionV3 models with accuracies of 92.4, 91.1, 92.3, and 92.8%, respectively, for the same cluster size.

The classification performance of 95.1% was achieved with an SVM classifier integrated to the GoogLeNet features of the BoDHF on cluster size 8, which is highest in comparison to the other DNN models for the same classifier and cluster size. The AlexNet, ResNet-50, ResNet-101, and InceptionV3 models achieved a 92.0%, 92.8%, 91.5%, and 92.8% classification accuracy, respectively, for cluster size 8. The classification performance of the proposed model BoHDF when tested on various combinations of the integrated DNN feature extraction model and classifier has provided the highest performance over cluster size 8. Therefore, we have selected eight clusters in the proposed BoDHF model.

## 5. Conclusions

An EEG-based emotion recognition framework was proposed and evaluated on the DEAP and SEED datasets. Out of the 64 channels, only 32 of them were considered to reduce the cost and complexity. A hybrid combination of the handcrafted OMTLBP_SMC and the GoogLeNet DNN model was used to extract the features from the 2D spectrogram representing the EEG signals. Furthermore, the bag-of-features model with a cluster of size 8 was used to reduce the dimension of the developed hybrid model. The proposed model achieved 93.83% and 96.95% recognition accuracy on the DEAP and SEED databases, respectively, which was higher than recently reported works.

The limitation of the proposed BoDHF is its added complexity due to the feature extraction procedure comprising raw hybrid deep and handcrafted features and the vocabulary estimation for dimension reduction and adding robustness to the model. The future work in this project will include the development of merging the handcrafted layers in the DNN model to reduce the complexity.

## Figures and Tables

**Figure 1 sensors-23-00498-f001:**
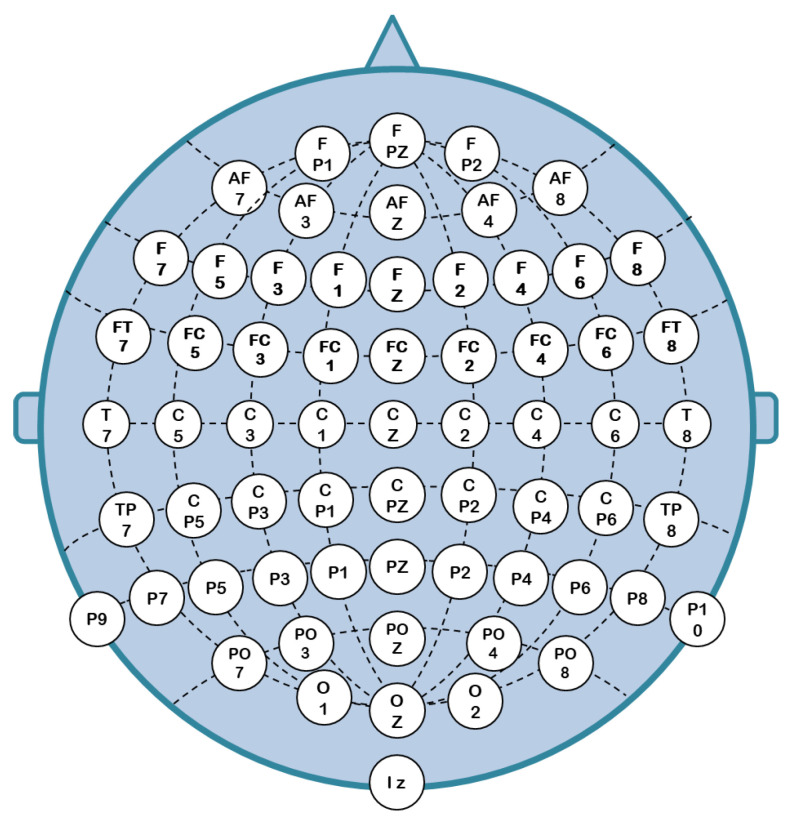
EEG electrodes placement on the human scalp.

**Figure 2 sensors-23-00498-f002:**
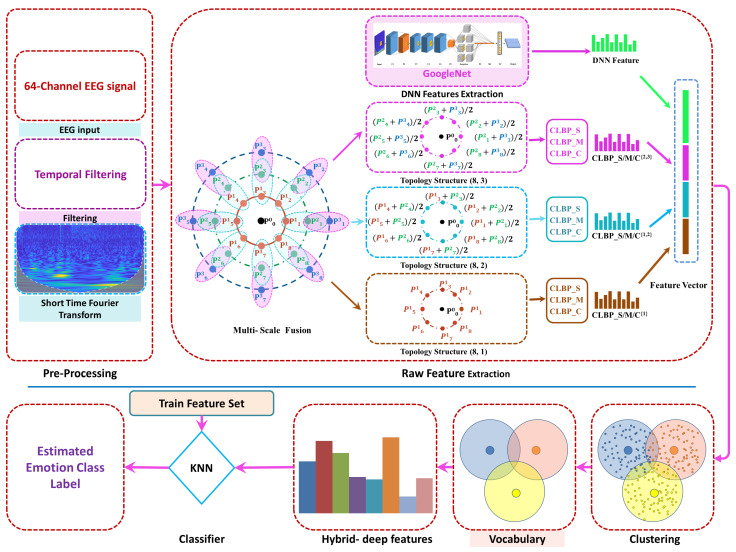
Proposed workflow.

**Figure 3 sensors-23-00498-f003:**
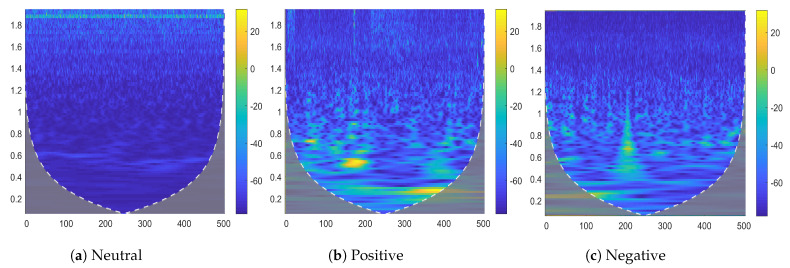
Two-dimensional spectrogram representing different emotions from the SEED dataset (Channel 1).

**Table 1 sensors-23-00498-t001:** Summary of frequently used techniques for emotion classification using EEG patterns.

Pre-Processing	Feature Extraction	Feature Reduction	Classification
Short-Time Fourier Transform	Power Spectral Density of Alpha, Beta, Gamma, Theta, and Delta	Maximum Redundancy Maximum Relevance (mRMR)	Convolutional Neural Network (CNN)
Down-sampling	Entropy	Linear Discriminant Analysis (LDA)	Deep Neural Network (DNN) (different variants)
Band-pass Filtering	Mean, Variance, and Standard Deviation	Principal Component Analysis (PCA)	Recurrent Neural Network (RNN)
Adaptive Filtering	Slope Sign Changes	Correlation-based Feature Selector (CFS)	Generative Adversarial Network (GAN)
Channel Selection	Hilbert–Hung Transform	Non-Negative Matrix Factorization (NNMF) and Others	Support Vector Machine (SVM)
Spectrogram Conversion	Power of DWT of Sub-bands	-	K- Nearest Neighbor (KNN)
Wavelet Transform	Recursive Energy Efficiency of DW	-	Naive Bayes (NB)
Discrete Cosine Transform and Others	High-Order Crossing and Others		Decision Tree (DT) and Others

**Table 2 sensors-23-00498-t002:** Publicly available databases for emotion identification.

Dataset	Year	Subjects (M/F)	Age Range	Stimuli	Acquisition Device
GAMEEMO [67]	2020	28	(20–27)	Emotional game video	EPOC+14 channel Emotiv
K-EmoCon [68]	2020	32 (20/12)	(19–36)	Video footage of the debate	NeuroSky Mind Wave Headset
SEED [69]	2017	15 (7/8)	-	Film clips	64-channels device
ASCERTAIN [70]	2016	58 (37/21)	Avg. 30	36 movie clips	Dry Electrode EEG device
DEAP [26]	2012	32 (16/16)	(19–37)	Video	Biosemi Active Two
MAHNOB-HCI [71]	2011	27 (11/16)	(19–40)	Video, images	Biosemi Active Two
IADS [72]	2007	-	-	Sounds	-

**Table 3 sensors-23-00498-t003:** Comparison of our proposed model with recent emotion identification methods.

Ref.	Method	Dataset	Channels	Classifier	Accuracy (%)
[64]	MFM	DEAP	18	CapsNet	68.2
[75]	MEMD	DEAP	12	ANN	75
[75]	MEMD	DEAP	12	KNN	67
[76]	MFCC	SEED	12	SVM	83.5
[77]	RFE	DEAP	12	SVM	60.5
[58]	DE	DEAP	32	PNN	79.3
[60]	BODF	DEAP	32	SVM	77.4
[60]	BODF	DEAP	32	KNN	73.6
[78]	EMD	DEAP	62	SVM	63.4
[78]	EMD	DEAP	62	KNN	57.3
[79]	Shannon’s entropy	DEAP	-	Multi-class SVM	94.09
[47]	EMD + sample entropy	DEAP	-	SVM	94.98
[47]	EMD + sample entropy	DEAP	-	Multi-class SVM	93.2
Proposed alg.	AlexNet	DEAP	28	SVM (Fine Gaussian)	92.1
Proposed alg.	AlexNet	DEAP	28	KNN (Weighted)	93.2
Proposed alg.	ResNet-50	DEAP	40	SVM (Fine Gaussian)	91.5
Proposed alg.	ResNet-50	DEAP	40	KNN (Weighted)	92.7
Proposed alg.	ResNet-101	DEAP	29	SVM (Fine Gaussian)	91.8
Proposed alg.	ResNet-101	DEAP	29	KNN (Weighted)	91.5
Proposed alg.	InceptionV3	DEAP	32	SVM (Cubic)	91.9
Proposed alg.	InceptionV3	DEAP	32	KNN (Weighted)	92.4
**Proposed alg.**	**GoogLeNet**	**DEAP**	**32**	**KNN (Weighted)**	**93.83**
[76]	MFCC	SEED	12	Random Forest	72.07
[65]	STRNN	SEED	62	CNN	89.5
[77]	RFE	SEED	18	SVM	90.4
[60]	BODF	SEED	62	SVM	93.8
[60]	BODF	SEED	62	KNN	91.4
[80]	HOS	SEED	34	SVM	82.1
[78]	EMD	SEED	62	SVM	56
[78]	EMD	SEED	62	KNN	52.6
Proposed alg.	AlexNet	SEED	28	SVM (Fine Gaussian)	93.7
Proposed alg.	AlexNet	SEED	28	KNN (Weighted)	94.6
Proposed alg.	ResNet-50	SEED	40	SVM (Fine Gaussian)	92.8
Proposed alg.	ResNet-50	SEED	40	KNN (Weighted)	93.9
Proposed alg.	ResNet-101	SEED	29	SVM (Fine Gaussian)	92.4
Proposed alg.	ResNet-101	SEED	29	KNN (Weighted)	92.8
Proposed alg.	InceptionV3	SEED	32	SVM (Cubic)	92.7
Proposed alg.	InceptionV3	SEED	32	KNN (Weighted)	93.3
**Proposed alg.**	**GoogLeNet**	**SEED**	**32**	**KNN (Weighted)**	**96.95**

**Table 4 sensors-23-00498-t004:** Performance comparison of the proposed method integrated with various DNN models and different cluster sizes using the SEED dataset.

Clusters	DNN Model	Classifier	Accuracy (%)	Clusters	DNN Model	Classifier	Accuracy (%)	Clusters	DNN Model	Classifier	Accuracy (%)
12	AlexNet	Ensemble	93.7	10	AlexNet	Ensemble	92.4	8	AlexNet	Ensemble	94.2
12	AlexNet	Tree	92.4	10	AlexNet	Tree	92.0	8	AlexNet	Tree	92.1
12	AlexNet	SVM	93.3	10	AlexNet	SVM	93.3	8	AlexNet	SVM	92.0
12	AlexNet	KNN	94.6	10	AlexNet	KNN	93.7	8	AlexNet	KNN	92.4
12	ResNet-50	Ensemble	93.7	10	ResNet-50	Ensemble	92.4	8	ResNet-50	Ensemble	93.3
12	ResNet-50	Tree	91.1	10	ResNet-50	Tree	90.2	8	ResNet-50	Tree	91.1
12	ResNet-50	SVM	92.8	10	ResNet-50	SVM	92.0	8	ResNet-50	SVM	92.8
12	ResNet-50	KNN	92.8	10	ResNet-50	KNN	92.4	8	ResNet-50	KNN	92.4
12	ResNet-101	Ensemble	92.4	10	ResNet-101	Ensemble	92.8	8	ResNet-101	Ensemble	93.2
12	ResNet-101	Tree	92.3	10	ResNet-101	Tree	92.0	8	ResNet-101	Tree	92.1
12	ResNet-101	SVM	92.0	10	ResNet-101	SVM	92.4	8	ResNet-101	SVM	91.5
12	ResNet-101	KNN	92.0	10	ResNet-101	KNN	92.4	8	ResNet-101	KNN	92.8
12	InceptionV3	Ensemble	94.2	10	InceptionV3	Ensemble	92.0	8	InceptionV3	Ensemble	93.5
12	InceptionV3	Tree	92.8	10	InceptionV3	Tree	94.3	8	InceptionV3	Tree	91.5
12	InceptionV3	SVM	92.4	10	InceptionV3	SVM	92.4	8	InceptionV3	SVM	92.8
12	InceptionV3	KNN	92.4	10	InceptionV3	KNN	92.0	8	InceptionV3	KNN	91.5
12	GoogLeNet	Ensemble	92.0	10	GoogLeNet	Ensemble	93.3	8	GoogLeNet	Ensemble	94.6
12	GoogLeNet	Tree	94.2	10	GoogLeNet	Tree	91.1	8	GoogLeNet	Tree	92.6
12	GoogLeNet	SVM	92.1	10	GoogLeNet	SVM	92.0	8	GoogLeNet	SVM	95.1
12	GoogLeNet	KNN	92.7	10	GoogLeNet	KNN	93.6	**8**	**GoogLeNet**	**KNN**	**96.95**

## Data Availability

The dataset utilized in this work is already availbie on the physio net website with titles DEAP and SEEDs, where the details have already been provided.

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
