# Peer review of "An AI-Inspired Spatio-Temporal Neural Network for EEG-Based Emotional Status"

_sensors, 2023, doi:10.3390/s23010498_

Round 1

Reviewer 1 Report

The topic of this paper is interesting, but the current version has many issues which have to improve in the revised version. 

The corrections are as follows:

1- In the abstract section, the research gaps are not clearly explained.

2- In the introduction section, the main contributions of this paper must be revised because some of them cannot be considered contributions. 

3- There are some typos and errors in grammar, the whole paper must be proofread by native English speakers.

4-  There are some duplications in the subsection that must be revised such as 2.1 and 4.1 both of them under "Data Pre-processing".

5- The result section must be revised and presented it in a more professional way to be more readable. 

6- The conclusion section is too short and must be revised.

7- What are the limitations of the proposed method?

Author Response

Thank you very much for your encouraging comment. We have answered, edited, and adjusted our manuscript according to your queries and suggestions.

Reviewer 2 Report

In the manuscript, a Bag-of-Hybrid Deep Feature (BoHDF) extraction model is proposed to classify EEG signals into respective emotion classes. STFT is used as preprocess to transform EEG signals into 2D spectrograms. The proposed methods combine the deep features from the GoogLeNet fully connected layer with texture-based features from OMTLBP_SMC and use K-nearest neighbor (KNN) for classification. The proposed method achieves high accuracy on the DEAP and SEED databases.

The quality of the manuscript does not meet the requirements of the journal now. Here are some questions and recommendations.

1. The structure of the manuscript is not appropriate. The Introduction and Related Work are too long. They should be streamlined and more content should be added to Experimental Results and Discussion. The main content has 13 pages, but the dataset and proposed method start on page 7. We suggest the authors to refer the structure of the published articles in this journal to make the manuscript more reasonable when submitting the paper. The existing work is not the core of the paper.

2. The authors should pay more attention to some details. For example, Table 1 could add corresponding references. Figure 3 lacks a description of the axes and color bar, and there is no description of the gray shading of the figure.

3. The description should be focused on the dataset used in the results section, rather than expressing irrelevance at great length. The followings need more description: the size of the dataset, the number of samples, and the size of each sample.

4. It will be better to explain the principle of OMTLBP_SMC and why it works for feature extraction, rather than just writing the calculation process and listing the formula.

5. The statement “The Bag-of-Hybrid Deep Feature (BoHDF) vector for any emotional status is extracted by calculating the frequency of vocabulary in the raw features of spectrograms.” is confusing. As it is said that the final raw feature vector is of size 1600, how to calculate the frequency of vocabulary?

6. Some sentences in the manuscripts are not grammatically correct. For examples,

“Among the different models used for extracting emotion-relevant features from EEG signals are the Deep Neural Network (DNN) models which fit well with neural processes in the brain.” in the abstract.

“P denote the overlapped multi-oriented pixel fusion obtained from the mean value of the amplitude values of the spectrogram segment laying at radius 2 and 3 are shown in Eq. 3 and 4.” in Section 4.2

7. The result analysis is insufficient. There is no analysis to explain why the proposed method can obtain higher accuracy compared to other methods.

8. Why were only k = 8, 10, 12 considered in the experiment to find the optimal value of k? How to explain that k = 8 achieves better results?

9. Ablation experiments need to be more fully described, and there are no results to support the benefits of STFT, which is mentioned to be related to “Spatio-Temporal” in the title.

Author Response

(The authors gave the same response as above.)

Round 2

Reviewer 1 Report

The authors have addressed all my comments, no more comments from my side. 

Reviewer 2 Report

Authors have addressed my questions.